# Optimizing Gradation Design for Ultra-Thin Wearing Course Asphalt

**DOI:** 10.3390/ma13010189

**Published:** 2020-01-02

**Authors:** Wentian Cui, Kuanghuai Wu, Xu Cai, Haizhu Tang, Wenke Huang

**Affiliations:** School of Civil Engineering, Guangzhou University, Guangzhou 510006, China

**Keywords:** functional pavement layer, ultra-thin wearing course asphalt mixture, high-temperature stability, pavement performance test

## Abstract

In recent years, ultra-thin wearing course asphalt mixture has been widely used in the reconstruction of old road surfaces and the functional layer of new road surfaces due to its good road performance. To improve the rutting resistance of ultra-thin wearing course asphalt mixture, this research presents an Ultra-thin Wearing Course-10 (UTWC-10) asphalt mixture with good high-temperature stability and skid resistance based on the Taylor system standard mesh specifications. The Course Aggregate Void Filling (CAVF) method is used to design the UTWC-10 asphalt mixture, which is compared with two other traditional ultra-thin wearing course asphalt mixtures on the basis of different laboratory performance tests. The high-temperature rutting test data shows that the rutting dynamic stability (DS) index of the UTWC-10 asphalt mixture is much higher than that of traditional wearing course asphalt mixtures, as it has better high-temperature stability. Moreover, anti-sliding performance attenuation tests are conducted by a coarse aggregate polishing machine. The wear test results show that the skid resistance of the UTWC-10 asphalt mixture is promising. The anti-sliding performance attenuation test can effectively reflect the skid resistance attenuation trend of asphalt pavement at the long-term vehicle load. It is verified that the designed UTWC-10 asphalt mixture shows excellent high-temperature rutting resistance and skid resistance, as well as better low temperature crack resistance and water stability than the traditional wearing course asphalt mixtures.

## 1. Introduction

As the economy continues to grow and urbanization deepens, the traffic load is also rising year by year [1,2]. Most of the roads will suffer from defects such as cracking, loosening, and deformation after 3–5 years of service, which will sharply weaken important pavement performance factors like water damage resistance, high-temperature stability, and skid resistance. These defects will not only shorten the service life of roads but will also bring severe threats to comfort and safety [3]. To reduce the occurrence of these pavement defects, researchers have started to study preventive pavement maintenance technologies [4]. The ultra-thin wearing course asphalt mixture can enhance the whole flatness and skid resistance of roads, reduce pavement noise, improve driving comfort and safety, prolong the time until surface cracking, fix local potholes, and other defects on pavement occur, as well as enhance the service life of roads [5,6,7]. Therefore, it has been widely used in the pavement rehabilitation for the extension of the service life of roads. 

As an overlay, the ultra-thin wearing course is the uppermost layer in pavement structures. In case of insufficient rutting resistance of ultra-thin wearing course, defects such as waves, displacements, and ruts will occur after a repetitive traffic load, thus affecting the slip resistance and driving comfort of pavement and threatening the safety of driving. However, few studies of ultra-thin wearing course were focused on the high-temperature resistance to permanent deformation [8,9,10,11,12].

At present, coarse aggregate is mostly concentrated at 4.75–9.5 mm, when the gradation design of ultra-thin wearing course asphalt mixture is designed to be open- or semi-graded. However, because the distance between the 4.75 mm mesh hole and 9.5 mm mesh hole is slightly larger, the uniformity of the coarse aggregate cannot be controlled. This will reduce the contact points between coarse aggregates and affect the spatial structure of the overall framework of the asphalt mixture. As a result, the embedded extrusion force between coarse aggregates will be reduced, the dynamic stability (DS) of the ultra-thin wearing course will be low, and rutting resistance will be insufficient [13,14]. Therefore, 8 mm (two meshes) and 6.7 mm (three meshes) mesh holes are added between 4.75 mm (four meshes) and 9.5 mm according to the Taylor system standard mesh specifications [15]. 

This research presents a new gradation design method to improve the high-temperature performance of the Ultra-thin Wearing Course-10 (UTWC-10) asphalt mixture based on the Course Aggregate Void Filling (CAVF) designed method and the mechanical performances were evaluated with two other commonly used open-graded friction courses OGFC-7 and Novachip-B asphalt mixture.

## 2. Materials and Methods

### 2.1. Raw Materials

#### 2.1.1. Aggregates

Diabase, consisting of 5–10 mm macadam, 3–5 mm macadam, and 0–3 mm stone chips was used as the aggregate of asphalt mixture. Table 1 summarizes the basic performance test results. Tests were conducted according to the Chinese specifications [16].

#### 2.1.2. Asphalt Binder

Here, high viscosity asphalt was used as asphalt binder. Table 2 shows the test results obtained for basic rheological properties. Tests were conducted according to the Chinese specifications [17].

#### 2.1.3. Mineral Filler

The filler used was alkaline limestone mineral filler. The impurities in the mineral filler were removed. Table 3 summarizes the obtained performance test results. Tests were conducted according to the Chinese specifications [16].

### 2.2. Methods and Preparation

#### 2.2.1. Asphalt Mixture Gradation Design

In the design of the target mix proportion of ultra-thin wearing course UTWC-10, the skid resistance and high-temperature stability performance of asphalt mixture pavement need to be considered. The gradation curve can be more reasonably controlled by adding 6.7 mm mesh holes and 8 mm mesh holes between 4.75 mm mesh holes and 9.5 mm mesh holes, in accordance with the Taylor system standard mesh specifications. Based on the porosity, the Course Aggregate Void Filling (CAVF) method was used to design three asphalt mixtures of ultra-thin wearing course with different porosities, and UTWC-10 was compared with OGFC-7 and Novachip-B. UTWC-10 was used as an example to illustrate the application of the CAVF method in the gradation design of asphalt mixture of ultra-thin wearing course, and that in Novachip-B and OGFC-7 was similar. The gradation design of fine aggregate was designed by the Fowler series as shown in Equation (1). The design process is shown in Figure 1 and durability test process is summarized in Figure 2. According to the specifications [16], the gross bulk density $\rho_{b}$ of the coarse aggregate was 2.90 g/cm^3^, and the bulk density ρ of the compacted state was 1.72 g/cm^3^. Therefore, it can be calculated from Equation (2) that the ${\mathrm{VCA}}_{\mathrm{DRC}}$ was 40.7%. The amount of mineral filler (*q_p_*), the target void ratio (*V_v_*) the asphalt content (*q_a_*) were 5%, 12%, and 5%, respectively. The amounts of coarse and fine aggregate were calculated using Equations (3) and (4) [18]:(1)$$P={\left(\frac{d}{D}\right)}^{n}$$
(2)$$VCA_{DRC}=\left(1-\frac{\rho }{\rho_{b}}\right)\times 100$$
(3)$$q_{c}+q_{f}+q_{p}=100$$
(4)$$\frac{q_{c}\times \left(VCA_{DRC}-Vv\right)}{100\times \rho }=\frac{q_{f}}{\rho_{af}}+\frac{q_{p}}{\rho_{f}}+\frac{q_{a}}{\rho_{a}}$$
where, *P* is the percentage of aggregate passing through the mesh size (%); *d* is the mesh size in mm; *D* is the maximum particle size of the aggregate in mm; *n* is the index, 0.3 ≤ n ≤ 0.5; $\rho_{b}$ is the gross bulk density of the synthetic coarse aggregate in g/cm^3^; ρ is the density of accumulation under rammed state in g/cm^3^; $\rho_{af}$ is the apparent relative density of fine aggregate in g/cm^3^; $\rho_{a}$ is the relative density of asphalt in g/cm^3^; and $\rho_{f}$ is the relative density of mineral filler.

Then, $\rho_{af}$ = 2.93 g/cm^3^, $\rho_{a}$ = 1.023 g/cm^3^, and $\rho_{f}$ = 2.864 g/cm^3^ were substituted in Equation (4). The quantity of coarse aggregate (*q_c_*) was found to be 72.2%, and the quantity of fine aggregate (*q_f_*) was 22.8%. The gradation of coarse and fine aggregates was fitted to a curve. The mesh hole passing rate and gradation curves of UTWC-10, Novachip-B, and OGFC-7 were designed by the CAVF method. The gradation compositions of the three ultra-thin wearing courses are shown in Table 4, and the gradation curves are shown in Figure 3.

#### 2.2.2. Determination of the Optimal Asphalt Content in Asphalt Mixture of the Ultra-Thin Wearing Course

The optimal asphalt content of UTWC-10 and Novachip-B were determined using the asphalt film thickness and void ratio, where the optimal thickness was 10 µm [19] and the target porosity was 12%. The optimal asphalt content of OGFC-7 was determined to be 4.5% through a leakage test and scattering test. 

The molded five groups’ asphalt content of asphalt mixture are 4.4%, 4.7%, 5.0%, 5.3% and 5.6% respectively, with 0.3% spacing distance between each group. Then, the optimum asphalt content is confirmed to be 5.0% for Novachip-B and UTWC-10 according to bulk density, void fraction, VMA, VFA, asphalt membrane thickness test data, and technical requirements of corresponding test.

The optimal asphalt content of OGFC-7 was determined through leakage test and scattering test. The five groups asphalt mixture with asphalt content to be 3.9%, 4.2%, 4.5%, 4.8%, and 5.1% shall be stirred. Marshall compaction instrument is used to mold the test specimen. It compacts 25 times on both sides. The formed specimen is put in the Los Angles abrasion machine to turn 300 circles at a speed of 33 r/min. Then, the ratio of lost weight and the original weight of the specimen is the flying loss. When the asphalt content of the open graded friction course OGFC-7 is between 4.4% and 4.6%, the change rate in leakage analysis loss and flying loss varies greatly. And since both the two losses of the asphalt mixture at this time meet the specification requirement, 4.5% is the optimum asphalt content of OGFC-7.

The volume parameters of the three asphalt mixtures under the optimum asphalt content are shown in Table 5, and the detailed volume parameters of the three asphalt mixtures can be seen in Table A1, Table A2, Table A3, Table A4 and Table A5 in Appendix A.

### 2.3. Methods and Tests

#### 2.3.1. High-Temperature Rutting Test

Rutting dynamic stability (DS) index can be used to check the asphalt texture stability at a high temperature. The larger the DS value, the better asphalt mixture performance in resistance to deformation and high temperature. Rutting modeling machine of asphalt mixture is used to mold the 300 mm × 300 mm × 50 mm rutting plate. The wheel tracking tests instrument is shown in Figure 4. Then, it is put for curing for 48 h. Rut test can be conducted after 5 h at temperature 60 °C. Wheel pressure and roundtrip speed of 0.7 MPa and 42 times/min were applied in tests. The wheel driving direction is consistent with the rolling direction in specimen molding. The deformation of asphalt mixture is 45 min, and 60 min is recorded separately. Total round trip times divides the gap of specimen deformation in 60 min and 45 min to gain the value of DS. DS can be calculated using Equation (5):(5)$$DS=\frac{\left(t_{2}-t_{1}\right)\times 42}{d_{2}-d_{1}}\times c_{1}\times c_{2}$$
where *d*_1_ and *d*_2_ are tracking depths at 45 and 60 min, *t*_1_ and *t*_2_ are 45 and 60 min, respectively; $c_{1}$ and $c_{2}$ are correction factors.

#### 2.3.2. Three-point Beam-bending Test

The low temperature crack resistance of asphalt mixture was evaluated by the small beam specimen bend test at low temperature. The formed rutting plate is cut to 250 mm × 30 mm × 35 mm trabecular specimen. Then, all specimens will be put in the incubator for about 6 h to enable the interior of specimen to reach the given temperature. The test temperature is –10 °C, 0 °C, and 15 °C. The specimen reaching the temperature shall be taken out to be put in the two-point support frame. The universal testing machine will load by means of mid-point loading at a speed of 50mm/min. The maximum bending strength and strain at failure were calculated and employed as evaluation indices for asphalt mixture low temperature crack resistance. Calculations were performed using Equations (6)–(8):(6)$$R=\frac{3LP}{2bh^{2}}$$
(7)$$\varepsilon =\frac{6hd}{L^{2}}$$
(8)$$S=\frac{R}{\varepsilon }$$
where *ε* is maximum bending strain at failure; *P* is breaking load (N); *R* is damage strength (MPa); *S* is stiffness modulus (MPa); *h* is cross-section sample height (mm); *L* sample length (mm); *d* is mid-span deflection at sample breaking point (mm); and *b* is test specimen width across middle section (mm).

#### 2.3.3. Immersed Marshall Test

The immersed Marshall method is quite simple and highly practical. The standard and formed Marshall test specimens are divided into two groups. One group is cured in 60 °C thermostatic water tank for 30 min, then goes for the Marshall stability test. The other group is cured for 48 h in 60 °C thermostatic water tank, then goes for the stability test. The ratio of the two stability values is residual stability. The closer the residual stability is to 1, the better asphalt mixture water stability. Calculations were performed based on Equation (9):(9)$$MS=\frac{S_{2}}{S_{1}}$$
where *MS* is test specimen residual stability (%); *S*_1_ is stability after immersing test specimen in water for 30 min (kN); and *S*_2_ is stability after immersing test specimen in water for 48 h (kN).

#### 2.3.4. Freeze-Thaw Splitting Test

The Marshall test method is adopted to mold two test specimen groups. One group is immersed in 25 °C water bath for 2 h, then its splitting strength (R_1_) is tested. The other group is immersed for vacuum treatment and water saturation for 15 min at 98 kPa, then the vacuum valve is switched on to cure for 30 min in water bath in ordinary pressure. 10 mL water shall be poured after the vacuum and water-saturated test specimen is sealed with a plastic bag. Next, it is tightened, sealed, and put in the −18 °C incubator to cure for 16 h. After that, the specimen saved in low temperature is taken out from the incubator and put in the 60 °C thermostatic water tank to cure for 24 h. At last, the test specimen is taken out from the hot water bath and put in 25 °C water. As can be seen from Figure 5, specimens were then removed to perform splitting tests according to the Chinese specifications [17]. After 2 h of constant temperature curing, its splitting strength (R_2_) is tested. The ratio of R_2_ and R_1_ is splitting tensile strength ratio (TSR). Calculations were performed using Equations (10) and (11):(10)$$R=0.006287\frac{P}{h}$$
(11)$$TSR=\frac{R_{2}}{R_{1}}$$
where *TSR* is freeze thaw splitting strength ratio; *R* is splitting tensile strength (MPa); *R*_1_ and *R*_2_ are average splitting tensile strength without and after freeze-thaw cycle (MPa), respectively; *P* is single specimen test load (N); and *h* is single specimen height (mm).

#### 2.3.5. Texture Depth Test

The sand patch method is commonly used in texture depth test, which is simple and widely applied. Firstly, 0.15 mm to 3 mm dry and clean sands shall be prepared. Then, they are filled into a 25 mL sand measuring cylinder, knock on the cylinder and bulldoze the cylinder mouth. After the surface of the rutting plate specimen is cleaned, fine sands are poured into the cylinder slowly. Then, the sands are spread outward to a circle with a push plate as much as possible and filled in the interspace of the test specimen. Lastly, the diameter of the circle in the two vertical directions can be measured and its average value shall be gained. The calculation method is shown in Equation (12):(12)$$TD=\frac{1000V}{\pi D^{2}/4}=\frac{31831}{D^{2}}$$
where *TD* is pavement texture depth (mm); *V* is sand volume (cm^3^); and *D* is paved sand average diameter (mm).

#### 2.3.6. British Pendulum Number (BPN) Test

The BPN test makes use of pendulum type friction coefficient measuring instrument (BPN tester) to get the BPN in bituminous pavement and cement concrete pavement in order to evaluate the anti-slide performance of the pavement in wet environments. The film in the bottom of the pendulum bob stands for the wheel. The pendulum bob falls from a certain height and the film in the bottom will rub for a certain distance on the pavement before swinging back. The height of swinging back is the pendulum of this section. In this article, the test specimen of the rut plate standards for tested pavement.

#### 2.3.7. Anti-Sliding Performance Attenuation Test

In order to simulate the actual paving thickness of ultra-thin wearing course, 1.5 cm thick rutting plate is molded according to the method described in Section 2.3.1. The rutting plate is cut to 30 mm × 1.5 mm × 8 mm slices and put in the steps of standard test film. Then, glue sands are made in the test of polishing value of coarse aggregate and filled in the interspace of the test film (Figure 6). After the test specimen demolds, it shall be put in the road wheel of the polishing machine. The rubber load and function times of the polishing machine shall be adjusted to test the BPN variance before and after abrasion. The value of BPN can be obtained by using pendulum type friction coefficient measuring instrument (BPN tester). The anti-slide performance attenuation acts as the main index to evaluate the anti-slide performance of asphalt mixture.

## 3. Results and Discussion

### 3.1. High-Temperature Rutting Resistance

The dynamic stability of the three asphalt mixtures meets the specification of being not less than 3000 times/mm [17]. For the UTWC-10 asphalt mixture shown in Table 6, the high-temperature stability was far more than that of OGFC-7, and the DS was 77.2% more than that of OGFC-7. Compared with Novachip-B, the high-temperature stability of UTWC-10 was strongly improved, and the DS was 36.9% higher than that of Novachip-B. The reason for the improvement in the high-temperature stability of the ultra-thin wearing course is that when UTWC-10 was optimized for gradation design, 8 mm (two meshes) and 6.7 mm (three meshes) mesh holes were added between 4.75 mm (four meshes) and 9.5 mm, according to the Taylor system standard mesh specifications. Coarse aggregates with a particle size of 4.75–6.7mm accounted for about 50% of the gradation. This increased the contact points between coarse aggregates, which made the spatial structure of the overall framework more reasonable and improved the embedded extrusion forces between coarse aggregates [20,21].

### 3.2. Low Temperature Crack Resistance

As can be seen from Figure 7, the stress and strain changes of the UTWC-10, Novachip-B, and OGFC-7 asphalt mixtures were similar. With temperatures from −10 °C to 15 °C, the bending strain of UTWC-10 increased by 34.5%, that of Novachip-B increased by 32.8%, and that of OGFC-7 increased by 12.9%. The bending strain of UTWC-10 increased the most with the increase in temperature. As the temperature rose, the ductility of the asphalt increased, improving the strain of the asphalt mixture. The bending strength of the three asphalt mixtures tended to be low with an increasing temperature, with a strength drop of 13.5% for UTWC-10, 14.3% for Novachip-B, and 28.5% for OGFC-7. The bending strength of UTWC-10 was least affected by temperature, because the rising temperature gradually increased the embedded squeeze forces within the framework of the UTWC-10 asphalt mixture. The bending stiffness modulus is the ratio of the bending strength and bending strain, and it is an important index for the evaluation of the low-temperature crack resistance of the asphalt mixture. The smaller the bending stiffness is, the better the elastoplasticity of the asphalt mixture is at the same damage strength. As the temperature increased, the bending stiffness modulus of UTWC-10 decreased by 35.7%, that of Novachip-B decreased by 35.4%, and that of OGFC-7 decreased by 36.7%. Because UTWC-10 had the fastest reduction in the bending stiffness modulus and the smallest bending stiffness modulus, it was shown to have the best crack resistance at low temperatures.

### 3.3. Water Stability

#### 3.3.1. Immersed Marshall Test Results

Figure 8 shows that the residual stability of the three asphalt mixtures meets the requirements of the specifications [17]. The residual stability of the three asphalt mixtures is above 90%, and the residual stability of UTWC-10 was 94.3%, that of Novachip-B was 92.9%, and that of OGFC-7 was 90.8%. Due to the high strength of the high viscosity asphalt used, the damage effect of the water immersion on the specimen is greatly reduced, so the residual stability of the OGFC-7 asphalt mixture with large void ratio was also good. The residual stability of utwc-10 is high and has good water stability. 

#### 3.3.2. Freeze-Thaw Splitting Test Results

It can be seen from Table 7 that the TSR value of UTWC-10 was 92.3%, the TSR value of Novachip-B was 90.4%, and the TSR value of OGFC-7 was 86.4%. Because the TSR value of UTWC-10 was the largest, the water stability performance of the UTWC-10 asphalt mixture was the best. The water porosity of the asphalt mixture will increase at low temperatures and produce tensile stress, causing micro cracks inside materials and thus attenuating the mechanical properties of the asphalt mixture. In the drainage material test of the OGFC-7 asphalt mixture, water was able to fill in most of the voids to reduce the free space of water after heaving. Therefore, the mechanical properties of heaved materials attenuated the fastest. UTWC-10 has a good framework strength and large density, and the heaving force of water has little effect on the mechanical properties of materials, so its freeze–thaw splitting strength ratio was shown to be the highest.

### 3.4. Asphalt Mixture Surface Roughness

Figure 9 shows that the texture depths of the three asphalt mixtures meet the requirements of the specifications [17] greater than or equal to 0.55 mm. The texture depth of UTWC-10 was shown to be 0.75 mm, that of Novachip-B was 0.73 mm, and that of OGFC-7 was 0.92 mm. OGFC-7 was shown to have the largest texture depth. The research results show that the void ratio is proportional to the texture depth index. The larger the void ratio is, the larger the texture depth is [22,23].

### 3.5. Anti-sliding Performance Attenuation

The anti-sliding performance attenuation test was completed by using a coarse aggregate polishing machine as shown in Figure 6. It can be seen from Figure 10 that as wear times rose, the pendulum BPN of the three asphalt mixtures showed a significant downward trend. The wearing results show that among the three ultra-thin wearing course, the attenuation rate of the anti-sliding performance of UTWC-10 was 15.80%, that of Novachip-B was 16.05%, and that of OGFC-7 was 18.18%. OGFC-7 was shown to have the highest anti-sliding performance attenuation rate. The attenuation rate of the anti-sliding performance of UTWC-10 and Novachip-B was similar. Besides, the BPN of Novachip-B was smaller than that of UTWC-10 and the skid resistance of UTWC-10 was the best. The anti-sliding performance attenuation test was able to effectively reflect the skid resistance attenuation trend of asphalt pavement at a long-term vehicle load.

## 4. Conclusions

This study proposed a UTWC-10 (Ultra-thin Wearing Course-10) asphalt mixture with good high-temperature stability and skid resistance. In the gradation design of UTWC-10, 8 mm (two meshes) and 6.7 mm (three meshes) mesh holes were added between 4.75 mm and 9.5 mm based on the Taylor system standard mesh specifications. Based on the laboratory tests and results discussion, the following conclusions can be drawn:(1)The DS of UTWC-10 asphalt mixture is as high as 5568 times/mm, which is much larger than Novachip-B and OGFC-7 asphalt mixture.(2)The test results of low temperature bending beam tests, immersed Marshall tests, and freeze–thaw splitting tests proved that UTWC-10 asphalt mixture has satisfied crack resistance at low temperatures and the ability to resist water damage.(3)The results of texture depth test and pendulum test confirmed that UTWC-10 asphalt mixture can provide good compactness and frictional resistance.(4)The anti-sliding performance attenuation test system employed in the paper can closely simulate the skid resistance attenuation of roads at a long-term vehicle load. The test results show that the skid resistance of UTWC-10 is the best.

## Figures and Tables

**Figure 1 materials-13-00189-f001:**
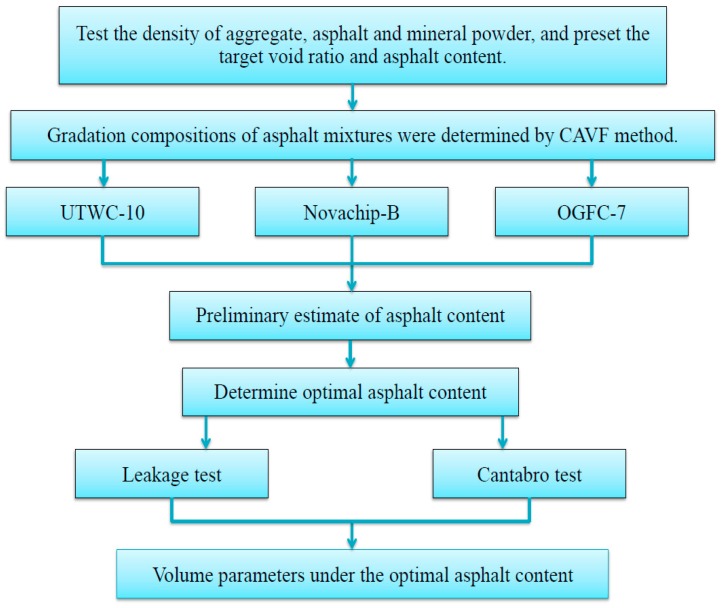
Design process of three asphalt mixtures.

**Figure 2 materials-13-00189-f002:**
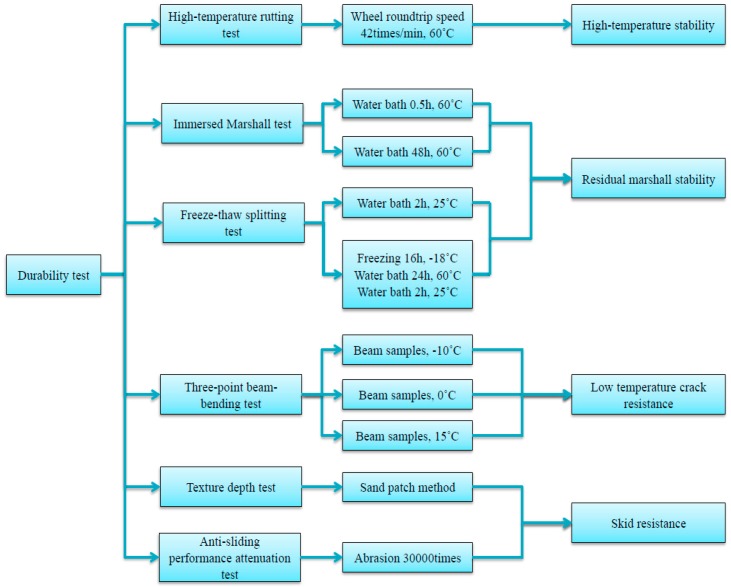
Flow chart of the durability test of three asphalt mixtures.

**Figure 3 materials-13-00189-f003:**
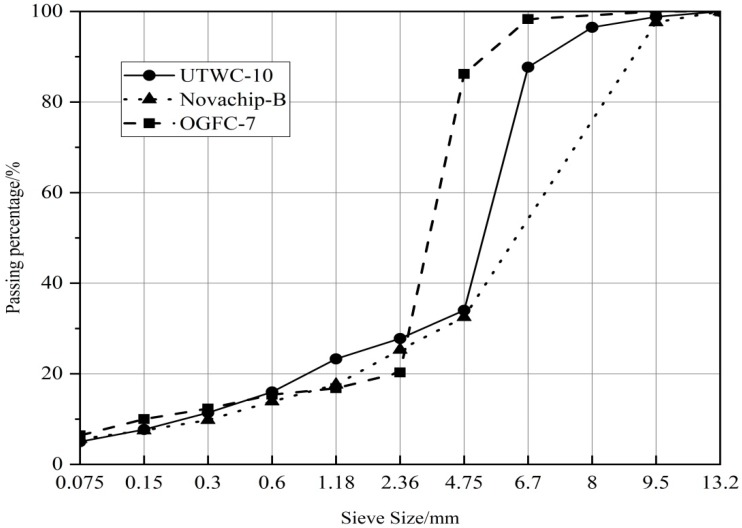
The gradation curves of the three asphalt mixtures.

**Figure 4 materials-13-00189-f004:**
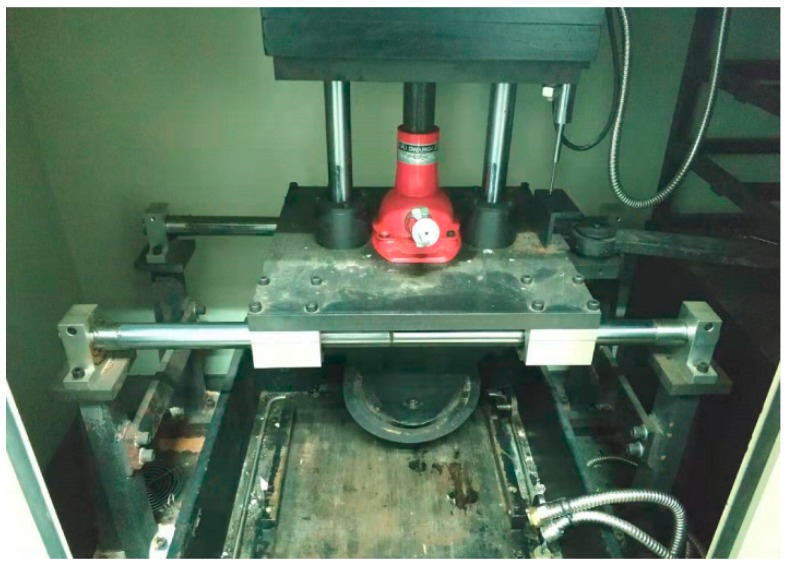
Wheel tracking tests instrument.

**Figure 5 materials-13-00189-f005:**
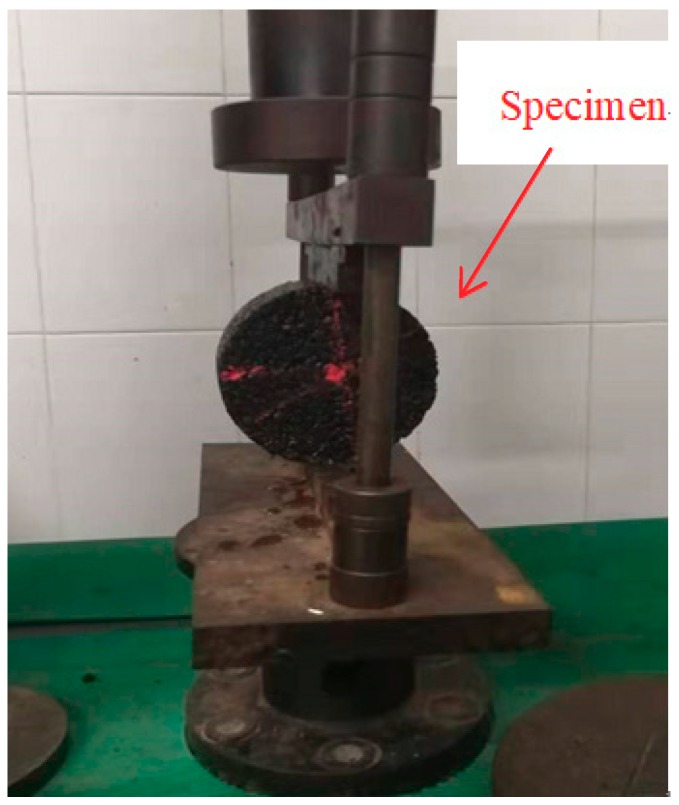
Freeze-thaw splitting test.

**Figure 6 materials-13-00189-f006:**
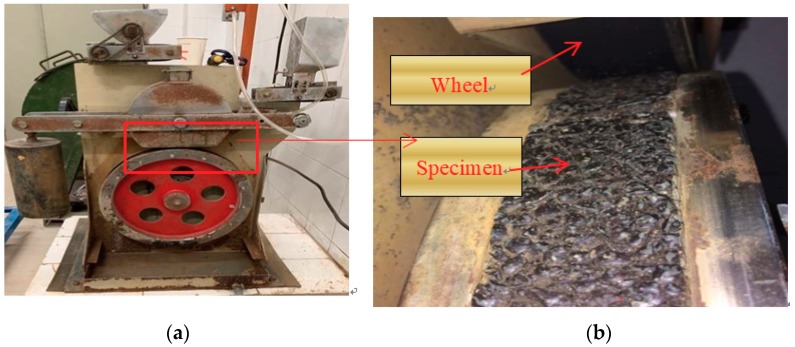
Wear test system: (**a**) coarse aggregate polishing machine; (**b**) ultra-thin wearing course specimen.

**Figure 7 materials-13-00189-f007:**
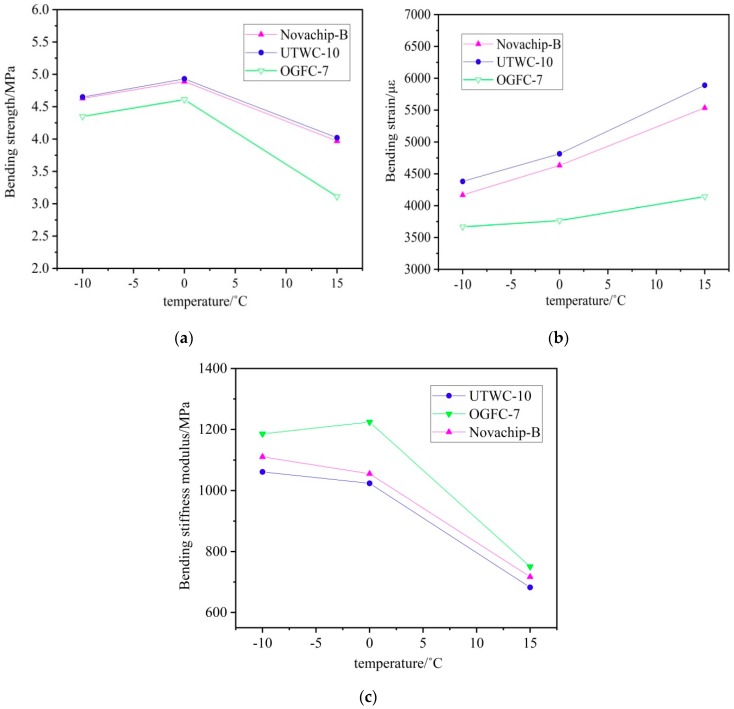
The various performance indexes of the three asphalt mixtures: (**a**) relationship between the bending strain and temperature; (**b**) relationship between the bending strength and temperature; and (**c**) relationship between the bending stiffness modulus and temperature.

**Figure 8 materials-13-00189-f008:**
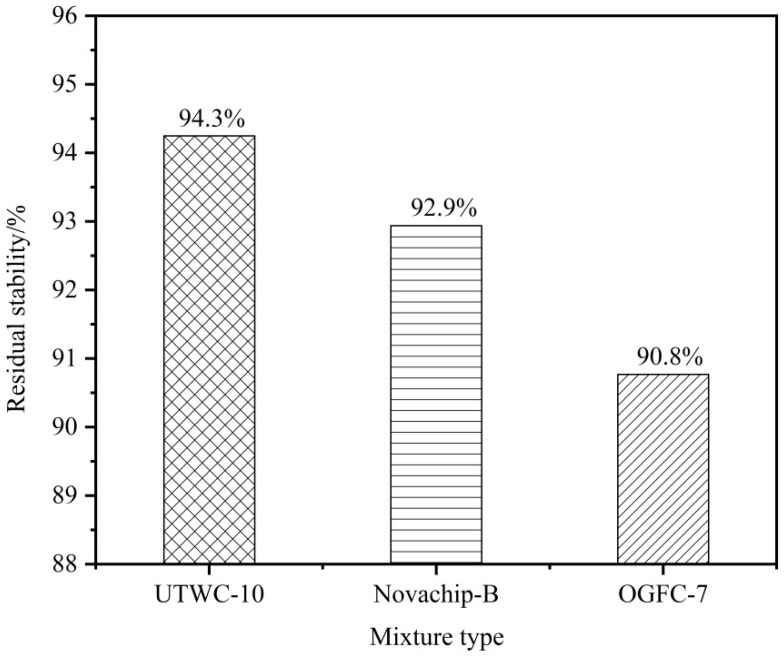
The average residual stability of the three asphalt mixtures.

**Figure 9 materials-13-00189-f009:**
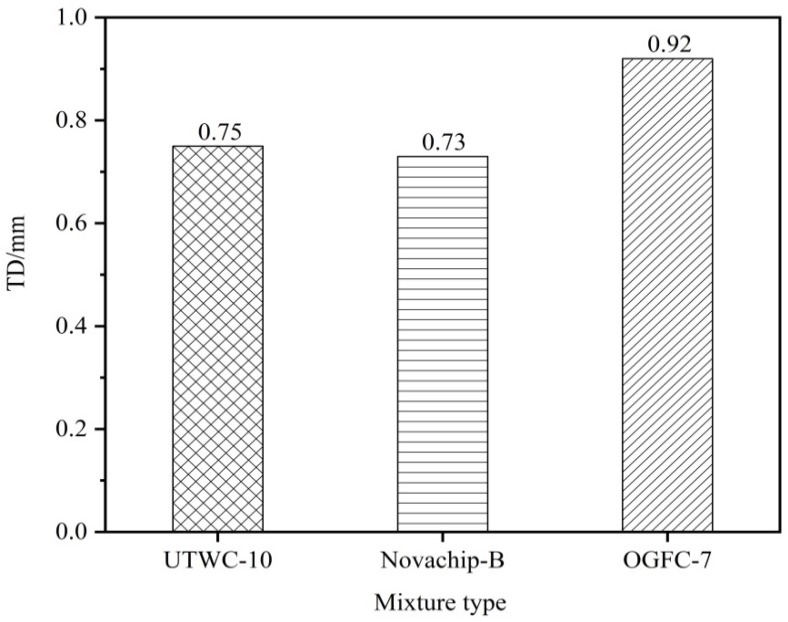
The texture depths of the three asphalt mixtures.

**Figure 10 materials-13-00189-f010:**
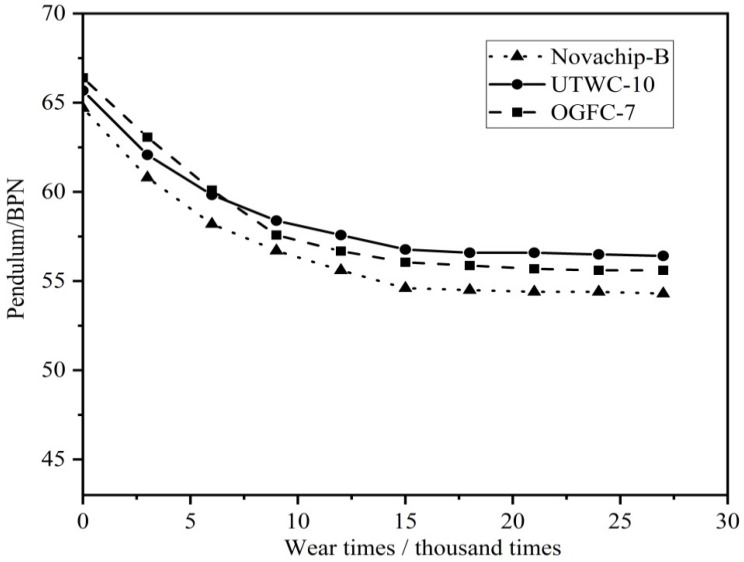
The anti-sliding performance attenuation curves of the three asphalt mixtures.

**Table 1 materials-13-00189-t001:** Properties of Aggregates.

Aggregate	Test Items	Standard Results	Test Results		
			5–10	3–5	0–3
Coarse aggregate	Crushed value (%)	≤26	4.9	5.3	-
	Abrasion value (%)	≤28	6.1	6.1	-
	Polished value PSV	≥42	49	-	-
	Needle shape (%)	≤15	3.2	-	-
	Water absorption rate (%)	≤2.0	0.87	0.96	-
	Gross bulk density (g/cm^3^)	-	2.90	2.89	-
	Apparent density (g/cm^3^)	>2.6	2.93	2.92	-
	Adhesion to asphalt (level)	≥5	5	-	-
Fine aggregate	Apparent density (g/cm^3^)	≥2.5	-	-	2.93
	Sand equivalent (%)	≥60	-	-	87
	Methylene blue value (g/kg)	≤25	-	-	14

**Table 2 materials-13-00189-t002:** Properties of asphalt binder.

Test Items	Unit	Standard Results	Test Results
Penetration index (25 °C)	0.1 mm	40–80	43.1
Softening Point T_R&B_	°C	≥70	81.6
Ductility (5 °C)	cm	≥20	29
Relative density of asphalt	g/cm^3^	-	1.023
Rotational viscosity 135 °C	Pa·s	-	2.958
Flexible recovery 25 °C	%	≥85	92
Rotating film oven test (163 °C)			
Quality loss	%	≤0.6	0.09
Penetration ratio	%	≥65	79
Ductility 5 °C	cm	≥15	20

**Table 3 materials-13-00189-t003:** Properties of mineral filler.

Test Items		Unit	Standard Results	Test Results
Apparent relative density		g/cm^3^	≥2.5	2.864
Hydrophilic coefficient		—	<1	0.6
Plasticity index		—	<4	1
Water content		%	≥1	0.08
Particle size range	<0.6 mm	%	100	100
	<0.15 mm	%	90–100	99.1
	<0.075 mm	%	70–100	97.7

**Table 4 materials-13-00189-t004:** The gradation compositions of the three asphalt mixtures.

Gradation Type	Passing Rate (%) under Different Mesh Apertures (mm)										
	13.2	9.5	8	6.7	4.75	2.36	1.18	0.6	0.3	0.15	0.075
Novachip-B	100	97.6	-	-	32.5	25.3	17.6	13.9	9.8	7.5	5.6
OGFC-7	100	100	-	98.3	86.2	20.3	16.8	15.4	12.3	10.0	6.4
UTWC-10	100	98.8	96.5	87.7	34.0	27.8	23.3	16	11.4	7.7	5

**Table 5 materials-13-00189-t005:** The volume parameters of thea three asphalt mixtures.

Asphalt Mixture Type	Optimal Asphalt Content (%)	Void Ratio (%)	VMA (%)	VFA (%)	Stability (kN)	Flow Value (mm)
UTWC-10	5.0	12.1	21.9	44.7	8.64	27.4
Novachip-B	5.0	12.3	22.4	43.5	8.43	32.1
OGFC-7	4.5	20.3	28.3	28.3	6.80	38.5

**Table 6 materials-13-00189-t006:** The high-temperature rutting test results of the three asphalt mixtures.

Gradation	45 min Deformation (mm)	60 min Deformation (mm)	Deformation Difference (mm)	DS (times/mm)	Mean Value of DS (times/mm)
UTWC-10	1.26	1.38	0.12	5250.00	5568
	1.30	1.41	0.11	5727.27	
	1.28	1.39	0.11	5727.27	
Novachip-B	1.47	1.63	0.16	3937.5	4025
	1.51	1.66	0.15	4200.00	
	1.49	1.65	0.16	3937.5	
OGFC-7	2.10	2.30	0.20	3150.00	3150
	2.13	2.33	0.21	2985.78	
	2.15	2.34	0.19	3314.22	

**Table 7 materials-13-00189-t007:** Freeze-thaw splitting test results of three asphalt mixtures.

Asphalt Mixture Type	Original Splitting Strength (R_1_)/MPa	Freeze-thaw Splitting Strength (R_2_)/MPa	Strength Ratio	Standard Results
UTWC-10	0.765	0.706	92.3%	≥80%
Novachip-B	0.757	0.684	90.4%	
OGFC-7	0.543	0.469	86.4%	

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
