# Peer review of "Optimizing Gradation Design for Ultra-Thin Wearing Course Asphalt"

_materials, 2020, doi:10.3390/ma13010189_

Round 1

Reviewer 1 Report

Thank you for made corrections.

Author Response

Dear Reviewers,

Thanks very much for taking your time to review this manuscript. I really appreciate all your comments and suggestions! Please find my itemized responses in below and my revisions in the re-submitted files.

Sincerely,

Wentian Cui

Point 1Thank you for made corrections.

Response 1: We are very appreciated the opportunity you given us. We revised the article carefully. We hope you can be satisfied with it.

Reviewer 2 Report

This is an interesting paper related to development of ultra thin asphalt pavement layer by changing aggregate gradation 1. Please do not mention specific company name. In worst case an unexpected conflict can be happened. 2. In Table 2, so only one asphalt binder was used? To verify the performance of given asphalt material at least 2 types of asphalt binder need to be considered. 3. Various test methods were considered than performed but most of them provide not crucial mechanical performances. Can author consider other mechanical test such as IDT, SCB test? 4. In case of data (i.e. results comparison) try numerical comparison method such as statistical analysis not mere visual comparison. 5. Why there is red marked letters (or sentences) in the manuscript? Some indications or authors' personal interest? 6. Based on simple performance test, the derived conclusions look normal however, more considerations such as mathematical modeling, numerical analysis (such as 2D, 3D FEM) works are needed to further verify findings in this study. I would kindly recommend publications however, proper (and/or major) revision works are strongly recommended to be published in this journal

Author Response

Dear Reviewers,

Thanks very much for taking your time to review this manuscript. I really appreciate all your comments and suggestions! Please find my itemized responses in below and my revisions in the re-submitted files.

Sincerely,

Wentian Cui

Response to Reviewer 2 Comments:

Point 1: Please do not mention specific company name. In worst case an unexpected conflict can be happened.

Response 1: Thank you for your suggestions. The specific company name has been removed. (Line74-75, page2、Line83-84, page3、Line88-89, page3)

Point 2: In Table 2, so only one asphalt binder was used? To verify the performance of given asphalt material at least 2 types of asphalt binder need to be considered.

Response 2: Thank you for your guidance. We strongly agree with you that it is not enough to choose one asphalt binder and evaluate its asphalt properties. We made a detailed analysis on the asphalt properties of several kinds of high viscosity asphalt binder and selected a kind of asphalt binder with better performance, and the test results of it are shown in Table 2. The asphalt binder, aggregates and mineral filler used in the three asphalt mixtures are the same.

Point 3: Various test methods were considered than performed but most of them provide not crucial mechanical performances. Can author consider other mechanical test such as IDT, SCB test?

Response 3: We are very appreciated with these suggestions. It is necessary to test the crack resistance of three asphalt mixtures. The test methods for crack resistance of asphalt mixtures include indirect tensile test (IDT), three-point beam-bending test and semicircle bending test (SCB) [1-2]. Therefore, the three-point beam-bending test has been used to evaluate the crack resistance of three asphalt mixtures in our paper (Line178-194, page8). Thank you very much for recommending IDT and SCB methods. In our later experiments, the crack resistance of asphalt mixture will be evaluated by semicircle bending test. Thanks again for your suggestions.

Reference

Minkyum, K.; Louay N. M. Density and SCB measured fracture resistance of temperature segregated asphalt mixtures. Int. J. Pavement. Eng. 2017, 10, 112–121. Luo, P.F. Research on the Asphalt Mixture Crack Test Methods and Evaluation Indexes Based on SCB. Master’s Thesis, Chang’an University, Xi’an, China, 2017.

Point 4: In case of data (i.e. results comparison) try numerical comparison method such as statistical analysis not mere visual comparison.

Response 4: Thank you for your suggestions. When we get accurate and reliable data through a large number of experiments, we make a better statistical analysis of the data, and all statistical results have been summarized in the table. Because there is much data, we use more intuitive figures to express the data (Line284-287, page12、Line300, page13、Line321, page14、Line334, page14), and the data is fully analyzed in the discussion section.

Point 5: Why there is red marked letters (or sentences) in the manuscript? Some indications or authors' personal interest?

Response 5: We are very sorry for the red marked sentences in the manuscript. Because this is the second time the manuscript has been resubmitted, the editor emphasizes that the changes must be highlighted in red. The sentences marked in red have been changed to black.

Point 6: Based on simple performance test, the derived conclusions look normal however, more considerations such as mathematical modeling, numerical analysis (such as 2D, 3D FEM) works are needed to further verify findings in this study.

Response 6: We are very appreciated with these suggestions. The results of this study will be better verified by modeling the ultra-thin wearing course asphalt mixture with finite element simulation and analysis, and this is the main content of our next paper. In this paper, the microstructural finite element model is reconstructed by means of CT scanning and stress film, and the model is used to do noise reduction and skid resistance numerical simulation.

Reviewer 3 Report

Interesting paper.

Please, rewrite the abstract. The reviewer believe that this can be more concise and contain fewer details that are not needed in this part of the paper. Please, focus on the important message to deliver. The body of the text appears to have different spacing. Please, check and correct. In the introduction, the literature review is interesting. However, this section is quite long. I would separate the research approach in a different section. Please, provide a flow chart of the research approach. Table 1 and 2 are hard to understand. Please, revise and use a graphical style that lead to less confusion. Why was a single (Ultra-thin Wearing Course-10) mixture selected? Can the author provides examples of other UTWC in the paper? Plots are sometimes not very clear. Fonts are small, numbers overlap with axes, etc. Please, revise. Table 5 is split across pages. Please, revise. It is recommended to include a picture of the testing device used for the rutting dynamic stability index DS. This will help the reader to immediately visualize the test used. Please, include also a picture for the Freeze-thaw splitting test. Please, improve the conclusions section. It is too long. Overall, English needs to be improved. The selection of words and the structures of sentences and paragraph is sometimes unusual. The paper is valuable from a practical point of view to engineers and practitioners.

Author Response

Dear Reviewers,

Thanks very much for taking your time to review this manuscript. I really appreciate all your comments and suggestions! Please find my itemized responses in below and my revisions in the re-submitted files.

Sincerely,

Wentian Cui

Response to Reviewer 3 Comments:

Point 1: Please, rewrite the abstract. The reviewer believe that this can be more concise and contain fewer details that are not needed in this part of the paper. Please, focus on the important message to deliver.

Response 1: Thank you for your suggestions. The abstract has been rewritten (Line7-23, page1). The new abstract is more concise, and it better highlights the pavement performance of UTWC-10 asphalt mixture.

Point 2: The body of the text appears to have different spacing. Please, check and correct.

Response 2: We are sorry for the different spacing. We have been corrected and checked it.

Point 3: In the introduction, the literature review is interesting. However, this section is quite long. I would separate the research approach in a different section. Please, provide a flow chart of the research approach.

Response 3: Thank you for your guidance. The flow charts of the research approach have been added. (Line102-105, page4-5)

Point 4: Table 1 and 2 are hard to understand. Please, revise and use a graphical style that lead to less confusion.

Response 4: We are very grateful for your suggestions. The format of the table has been adjusted and is much clearer than before. (Line92, page3、Line100-101, page4)

Point 5: Why was a single (Ultra-thin Wearing Course-10) mixture selected? Can the author provides examples of other UTWC in the paper?

Response 5: Thank you for this suggestion. It is not enough to select a single ultra-thin wearing course asphalt mixture (UTWC-10). In this paper, we compared the commonly used open-graded friction course OGFC-7 (maximum nominal particle size of aggregate is 6.7mm) and the traditional Novachip-B asphalt mixture to UTWC-10. OGFC-7 and Novachip-B are both ultra-thin wearing course asphalt mixtures with a thickness of 1.5-2cm. The asphalt binder, aggregates and mineral powder used in the three asphalt mixtures are the same, and their gradation compositions are different. Through the comparison of the three asphalt mixtures, it is verified that the UTWC-10 asphalt mixture after the gradation optimization is significantly improved in high temperature stability, and the skid resistance, low temperature crack resistance and water stability still meet the requirements of the specification.

Point 6: Plots are sometimes not very clear. Fonts are small, numbers overlap with axes, etc. Please, revise.

Response 6: We are very sorry that some plots are not very clear. Figure 3 has been revised (Line139, page6), and the other figures have been checked and modified (Line284-287, page12、Line300, page13、Line321, page14、Line334, page14).

Point 7: Table 5 is split across pages. Please, revise.

Response 7: Thank you for your suggestions. It has been modified. (Line161, page7)

Point 8: It is recommended to include a picture of the testing device used for the rutting dynamic stability index DS. This will help the reader to immediately visualize the test used. Please, include also a picture for the Freeze-thaw splitting test.

Response 8: We are very appreciated with these suggestions. The picture of Wheel tracking tests instrument has been added (Line176, page8). The picture for the freeze-thaw splitting test is shown in Line220, page9.

Point 9: Please, improve the conclusions section. It is too long.

Response 9: Thank you for your suggestions. The conclusions have been rewritten. (Line337-351, page14-15)

Round 2

Reviewer 3 Report

The authors addressed the comments

This manuscript is a resubmission of an earlier submission. The following is a list of the peer review reports and author responses from that submission.

Round 1

Reviewer 1 Report

This is well structured paper with interesting case study presented in an appropriate manner. Still, I have some minor comments.

Though, those are some commonly used abbreviations, please defined them within first mention (SMA, GTM, GSI…).

Page 5, Line 162, “…optimal thickness was 10 um” is this a typo?

Page 7, Lines 209-212: Can you please elaborate (discuss) in more details results of Immersed Marshall Test. Is the results of UTWC-10 highest residual stability value influenced only by its granulometry?

Page 9, Lines 238-247: Can you please elaborate (discuss) in more details results of Anti-Sliding Performance Attenuation Test. Is the best result of UTWC-10 skid resistance comparing to other tested mixtures result only of its granulometry?

Can you please add some details on test sample preparation, size and number?

Reviewer 2 Report

Dear authors

thank you for your paper. Unfortunately, I have a lot of important comments:

The literature survey should be rewritten. All of the references, except one, are from researchers from the same country (only three institutions including the author's university) + a couple of standards. This means a very weak literature survey which is not acceptable. It is certainly beneficial to check what researchers of other countries and universities have done in this field as well. Reference number 20 appears several times in the text but is not present in the list of references. Three of the references are master theses and not easy to find. If possible, include better references, or at least include a link to the repository. If these theses are written in Chinese, also include this information. Section 2: materials are discussed, but no methods. You should at least include a proper overview of the used methods (e.g. number of samples, references to international standards, selected temperature, ...) and use the well-known names for them, e.g. High-temperature rutting test sounds like you perform the standardized rutting test at higher temperatures? What is the Tectonic depth test? A lot of abbreviations in the text are not explained properly It is unclear how different the three mixtures really are. Did you use the same type of binder, aggregates, ...? Is Novachip a commercial mixture? If so, then I assume that you use a specific binder and type of aggregates? I am not convinced that the gradation curves are correct. You do not know how the gradation is between two sieves, as it is dependent on the used aggregates. You claim to have implemented a change in mixture gradation between 4.75 and 9.5 mm, but this is not certain. It would be interesting to check the actual gradation of the mixtures that were used (extract the bitumen and perform another sieving analysis). And even if this gradation is slightly different, is it really such a large improvement? Some improvements are claimed in the paper, but not supported by actual results or previous studies (e.g. improved the embedded extrusion forces ..., greater compactness and higher cohesion ...) I expect that multiple samples were tested (I know that this is required for water sensitivity), so I expect some kind of statistical analysis, showing at least the standard deviation. This might influence some of the conclusions (overlapping SD = no significant difference between the averages) Anti-sliding ... = performed using a British pendulum tester?? all the shown results are unclear as you have no idea which actual test was performed. Also consider to not only compare between the three mixtures, but compare to results from other researchers (also outside of China). Make sure to include minimum or maximum standard values

Overall, this looks like the work performed by some master students, but with a low degree of novelty, so I do not recommend this paper for publishing.

I uploaded a document containing all my comments. This also includes markings for phrases or words which should be checked.

Round 2

Reviewer 2 Report

Dear authors

thank you for your revised version.

Unfortunately, most of my remarks are still unresolved. In a lot of cases, you reply in your Word document to my comments, e.g. explaining a certain issue (like V-S method or OGFC-7 or Novachip), without actually implementing this in the paper itself. In most of those cases, I know what you meant, but by asking it, I tried to make clear that it was not included or clear in the paper itself. In other cases, you made it worse instead of better (e.g. TR&B ==> you made R&B subscript, but you removed the T) Another example is the subscript of Table 5.

Furthermore, the Materials and Methods section still needs a lot of work. In some cases you have added information regarding the used test methods or number of samples, but not in Section 2. This should not be included in the section where you discuss the results. It should be stated more clear that all mixtures use the same binder.

You mention results about the optimum bitumen content for the mixtures, so why are those results not included in more detail? I also asked for statistical analysis (minimum a standard deviation) which is still not included. You mention in some cases that 36 samples were tested, so show the standard deviation besides the mean. Same applies for water sensitivity where you take the average of 3-5 samples ==> it is easy to determine the SD + I also asked to show the actual indirect tensile test, and not only the ITSR. This is very important as it will show that there are no statistically relevant differences between your mixture and Novachip for a number of tests. You could also test this using more complicated statistical tests ... (based on the variances between the means) You mention some SD in your reply but did not include them in table 6 or other graphs ..

Some other aspects which still remain open:

- I propose a table where you clearly compare the composition of each of the used mixtures, not only the % passing. ==> if you sieved all separate fractions individually, then this should be made more clear. If you used existing fractions like 2-6 and 6-10, then such a table is needed

- Similar remark: How certain are you that the Novachip-B is graded like this in between those sieves (straight line)? This is completely dependent on the used granulates ... ==> I refer here to how the mixture is made in the plant, based on available granulates. They will never sieve all granulates according to different mesh sizes ...

- Apparently one of your mixtures only has 4.5 % of bitumen and a different gradation, so of course you will find other results for that mixture ...

- the literature survey is still considered as weak, and highly focused on China. I am unable to locate all references, such as 13, 14 an 19 (only available in Chinese?), [15] is not a correct way to refer to a standard

Finally, it seems as you focused on the comments from the PDF and not the main summarized comments added online. 

kind regards

Round 3

Reviewer 2 Report

Dear authors

thank you for the revised version of your paper entitled "Optimizing gradation design for ultra-thin wearing course asphalt".

Some of my comments, such as a more elaborate literature review and restructuring of section 2, were dealt with (partly), but overall I still value the novelty of the paper as low.

This is for me an important criterion which will be very difficult to meet with the current results.

Three mixtures are compared, of which one has a lower % of bitumen and a clearly different gradation curve, so (large) differences can be expected. The difference in the gradation curve between the Novachip and UTWC-10 is small, but not only limited to the investigated area (between 4.75 and 9.5 mm). Furthermore, I have tried to explain the importance or ask about the (un)certainty that the Novachip mixture indeed follows this curve in between 4.75 and 9.5 mm (the straight line). This is highly influenced by the used mixture of (available) granulates. The current change to Table 4 still gives no answer to this question or concern. In the improved mixture, you now have about 10% of stones with a size of 8 or 6.7 mm, but you have no idea how many you had in the original Novachip mixture ... Not all results should be clarified or explained by the differences on those sieve sizes.

The requested standard statistical analysis is only partly available and in some cases wrong (e.g. averages in Table 6? or missing SD for Figure 3).

Some further comments:
- check Figure 1: X-axis labels
- References to standards for Section 2 Methods
- Check the style of the formulas, in this version split over two lines?
- Line 91: “open- graded” remove space
- Line 93: Novachip-B(ultra…)  add space before the parenthesis.
- Line 196: d1 and d2 is => are
- Line 215: “P is the is” … Remove the second “is”.
- Line 341: was90.4% and was86.4%  add space between text and numbers.
- Line 384  … properties of that mixture were compared …
- Line 477: Profilfile ?
- Sometimes there is a space before your unit, like in Line 134: 13.2-9.5 mm. Sometimes the units are right after the number, like in Line 233: 98 KPa (also, normally, kiloPascal is shown kPa not KPa) Check the whole manuscript and be consistent.
- Same for your references. Example: Line 287 [17] is after the text without any space. However, line 35 is different. Check the whole manuscript and be consistent.
- Table 4: please check if the new version is actually better than the old one
- As you explained in Response 6, OGFC-7 has a different amount of bitumen, so the differences between that mixture and the other two are not due to different gradation but combination of different gradation and bitumen content. Better to state that difference in the introduction at line 94 when you say: “The asphalt binder, aggregate and mineral powder used in the three asphalt mixtures are the same.”
- Is OGFC on line 165 the same as OGFC-7 or another mixture? If they are the same, line 171 is repeating line 165.

kind regards
the reviewer